

# ResMem-Net: memory based deep CNN for image memorability estimation

Arockia Praveen[1], Abdulfattah Noorwali[2], Duraimurugan Samiayya[3], Mohammad Zubair Khan[4], Durai Raj Vincent P M[5], Ali Kashif Bashir[6] and Vinoth Alagupandi[7]

[1] Phosphene AI, Madurai, India
[2] Umm Al-Qura University, Makkah, Saudi Arabia
[3] Department of Information Technology, St. Joseph's College of Engineering, Chennai, India
[4] Department of Computer Science, Taibah University, Medina, Saudi Arabia
[5] School of Information Technology and Engineering, Vellore Institute of Technology, Vellore, Tamilnadu, India
[6] The Manchester Metropolitan University, Manchester, United Kingdom
[7] Optisol Business Solutions, Chennai, India

Corresponding author
Durai Raj Vincent P M,
pmvincent@vit.ac.in

## ABSTRACT

Image memorability is a very hard problem in image processing due to its subjective nature. But due to the introduction of Deep Learning and the large availability of data and GPUs, great strides have been made in predicting the memorability of an image. In this paper, we propose a novel deep learning architecture called ResMem-Net that is a hybrid of LSTM and CNN that uses information from the hidden layers of the CNN to compute the memorability score of an image. The intermediate layers are important for predicting the output because they contain information about the intrinsic properties of the image. The proposed architecture automatically learns visual emotions and saliency, shown by the heatmaps generated using the GradRAM technique. We have also used the heatmaps and results to analyze and answer one of the most important questions in image memorability: "What makes an image memorable?". The model is trained and evaluated using the publicly available Large-scale Image Memorability dataset (LaMem) from MIT. The results show that the model achieves a rank correlation of 0.679 and a mean squared error of 0.011, which is better than the current state-of-the-art models and is close to human consistency ($p = 0.68$). The proposed architecture also has a significantly low number of parameters compared to the state-of-the-art architecture, making it memory efficient and suitable for production.

## INTRODUCTION

Data is core essential component to almost every media platform in this digital era, starting from the television to social networks. Every media platform relies on content to engage their users. It provides a compulsion for these platforms to understand the exponentially growing data to serve the right content to their users. Since most of these platforms rely on visual data, concepts such as popularity, emotions, interestingness, aesthetics, and, most

importantly, memorability are very crucial in increasing viewership (*Kong et al., 2016*; *Celikkale, Erdem & Erdem, 2015*). In this paper, image memorability is the concept taken into consideration which is one of the most underexplored deep learning applications.

Human beings normally rely on visual memories to remember things and also will be able to identify and discriminate objects in real life. Human cognition to properly remember and forget visual data is crucial as it affects every form of our engagement with the external world (*Bainbridge, Dilks & Oliva, 2017*; *Schurgin, 2018*). However, not all humans remember the same visual information in a common manner (*Gretz & Huff, 2020*; *Rust & Mehrpour, 2020*). It is a long-standing question that neuroscientists have asked for years, and research is still underway to explain how exactly the cognitive processes in the brain encode and store certain information to retrieve that information when required properly. The human brain can encode intrinsic information about objects, events, words, and images after a single exposure to visual data (*Alves et al., 2020*; *Fukuda & Vogel, 2019*).

Image memorability is generally measured as the probability that a person will be able to identify a repeated photograph when he or she is presented with a stream of images (*Isola et al., 2011a*). By definition, image memorability is a subjective measure that approximately quantifies how a person can remember an image (*Isola et al., 2011b*). Cognitive psychologists have shown that more memorable images leave a larger trace of the brain's long-term memory *Broers & Busch, 2021*. However, the memorability of a certain image can slightly vary from person to person and depends on the person's context and previous experiences (*Bainbridge, 2020*). But this slight variation is fine because this allows us to make approximate predictions using computational methods.

Researchers have shown that, even though there exist slight variations, humans show a level of consistency when remembering the same kind of images with a very similar probability irrespective of the time delay (*Sommer et al., 2021*). This research has led to the inference that it is possible to measure an individual's probability of remembering an image. To measure the probability that a person will remember an image, the person is presented with a stream of images. This process is called the Visual Memorability Game (*Isola et al., 2011a*). The stream of images contained two kinds of images, targets, and fillers. The annotator is shown images one by one, where the image is displayed for 2.4 s. Between each target image multiple filler images are shown unbeknownst to the annotator. On a random manner, previously shown target images are repeatedly shown now and then. When each image is being shown, the annotator is asked to press a key in the keyboard if that annotator feels that the target image is being repeated in the stream. Based on this, then the percentage of times the annotator has correctly identified repeated target images will be checked and is annotated as the memorability of the target image from one annotator. The same set of images are shown to multiple more people in the same manner. So, the approximate memorability scores will be obtained for the same image from multiple people. Image memorability is a reflection of individual viewing the image, but however, the level of memorability of an image is quite similar across individuals most of the time (*Sommer et al., 2021*). So, since the memorability of the image is only going to slightly vary for most people, these approximate measures are taken as the ground truth memorability score. The slight difference between most deep learning datasets and image

memorability datasets is that, for each image, we will have multiple annotations, that is, multiple memorability scores, which is fine because most of them aren't going to vary that much.

As mentioned earlier, unlike more objective properties of images such as photo composition or image quality, image memorability cannot be objectively defined and hence might slightly vary from person to person. However, generally, humans agree with each other on certain common factors that tend to make an image more memorable despite this large variability. Factors like color harmony and object interestingness are generally agreed upon by people as factors that improve image memorability *Khosla et al., 2015*.

Few methods have been proposed (*Perera, Tal & Zelnik-Manor, 2019*; *Fajtl et al., 2018*; *Squalli-Houssaini et al., 2018*) to predict the memorability of an image using deep learning methods. Those methods either used handcrafted features or ensemble models to predict the memorability score. Ensemble models are hard to train, computationally expensive and are prone to overfitting (*Canchumuni, Emerick & Pacheco, 2019*). The overfit models normally do not perform well on different kinds of images that are not in the training set, while computationally expensive models are not suitable for deployment to real world on web servers or computers with low memory GPUs and real-world deep learning systems are heavily reliant on computers with GPUs. Methods that use handcrafted features along with machine learning models are not accurate on different kinds of images because is it extremely hard to handcraft a comprehensive amount of features that can span a wide distribution.

The idea of using data-driven strategies to predict image memorability was first introduced by *Isola et al. (2011a)*. The Visual Memorability Game was used to prepare the images in the Isola et al. dataset and annotate their respective memorability score. The game was run on Amazon Mechanical Turk, where users were presented with a stream of images with some images repeating on a random basis. The users were asked to press a key when they believe that the image displayed was already seen before. In the Isola et al. dataset, they have collected 2,222 images along with the annotated memorability scores. Since memorability can vary slightly from person to person, each image was shown to 78 participants on an average, when the annotators played the Visual Memorability Game. Each image being tagged more than once accounts for the slight variation in memorability among people. This also means that when the deep learning model is being trained, during each epoch, the model will be given the same image as input multiple times but with a slightly varying ground truth. When they analyzed the images and their memorability scores together, they have understood that the memorability of an image is highly related to a certain object and scene semantics such as 'Labelled Object Counts,' 'Labelled Object Areas' and 'Object Label Presences.' Also, when each image was segregated into scene categories, it was inferred that much of what contributes to the image's memorability score was from both the object and scene semantics. They have followed up on their work to understand the human-understandable visual attributes to understand memorability as a cognitive process. They have developed a deep learning model that can predict scene category of an image to with another deep learning model that predicts image memorability to understand

and identify a compact set of image properties that affect image memorability (*Lu et al., 2020*).

A new dataset, the Large-scale Image Memorability dataset (LaMem), which is publicly available, is a novel and diverse dataset with 60,000 images, each tagged with memorability score similar to the dataset by Isola et al. *Khosla et al., 2015* have used Convolutional Neural Networks (MemNet) to fine-tune deep features that outperform all other features by a large margin. The analysis made by the author on the responses of high-level Convolutional Neural Networks (CNN) layers shows which objects are positive. A new computational model based on an attention mechanism to predict image memorability based on deep learning was proposed. In this paper, the authors have shown that emotional bias affects the performance of the proposed algorithm due to the deep learning framework arousing negative pictures than positive or neutral pictures (*Baveye et al., 2016*).

*Squalli-Houssaini et al, (2018)* presented a hybrid CNN with Support Vector Regression (SVR) model trained on the LaMem dataset. The model achieved an average rank correlation of 0.64 across the validation sets. Based on the predictions, the correlation between interestingness and memorability was analyzed. The predictions were compared using the Flickr Interestingness API and the results showed that memorability did not correlate much with interestingness (*Squalli-Houssaini et al., 2018*).

Visual attention has a huge effect on image memorability *Fajtl et al., 2018*. However, very little work has been done on taking advantage of visual attention to predict image memorability. *Mancas & Le Meur (2013)* proposed a model that uses a new set of attention-driven features by identifying the link between image saliency and image memorability. The model achieved a 2% increase in performance from the existing models. It was also inferred that images with highly localized regions are more memorable than those with specific regions of interest (*Mancas & Le Meur, 2013*).

A novel deep learning architecture was proposed that took advantage of the visual attention mechanism to predict image memorability by *Fajtl et al. (2018)*. The architecture made use of a hybrid of Feedforward CNN architecture and attention mechanism to build a model that can help build attention maps and, in turn, predict memorability scores. The model attained excellent results, but the biggest downside was overfitting and lack of the provision to use transfer learning swiftly. The model also contains a large number of parameters making it hard for real-time production.

Another model that used visual attention mechanism was proposed by *Zhu et al. (2020)*. The architecture is a multi-task learning network that was trained on LaMem dataset and AADB dataset (*Kong et al., 2016*) to predict both the memorability score and aesthetic score of an image, hence it was also trained using two datasets at the same time, one for image memorability and the other for image aesthetics. The model used a pixelwise contextual attention mechanism to generate feature maps. Even though this model was able to use transfer learning, the attention mechanism used is computationally expensive, especially if the number of channels in the intermediate layers is high. This model for the memorability task achieved a rank correlation of only 0.660, which is a much lower score than the ones achieved other existing models.

An ensemble model that predicts video memorability was proposed by *Zhao et al. (2021)*. The model was trained on the MediaEval2020 dataset and is an ensemble of models that extract audio, video, image and text features from the input to predict video memorability. The features of the audio were extracted using a pretrained VGG model, while the image and video features were extracted using a ResNet-152. These features were then passed onto other machine learning models to get the memorability score. It was found that Bayesian Ridge Regressor worked best for processing audio features while a Support Vector Regressor worked best for processing image and video features. The text features for the tagged human annotated captions were obtained using GloVe word embeddings. The model achieved a rank correlation of 0.370 for short term memorability and 0.289 for long term memorability on the validation set of the dataset.

A multi-modal fusion-based model trained on the MediaEval2019 dataset for video memorability prediction was proposed by *Leyva & Sanchez (2021)*. This model takes advantage of motion estimation techniques and combines it with text, audio and image features. To estimate motion and obtain its feature vectors, two 3DResNets were used. The image features were extracted using ResNet-56 and ResNet-152, while the text features were obtained using a combination of CNN and Gated Recurrent Unit (GRU). The feature vectors from text, image, and motion estimation are then processed through late fusion and then a Bayesian Ridge Regressor predicts the memorability score. On the validation set, the model obtained a rank correlation of 0.5577 for short term memorability and 0.3443 for long term memorability.

A Hidden Markov Model (HMM) produced using Variational Hierarchical Expectation Maximization was proposed by *Ellahi, Vigier & Callet (2020)*. A new dataset with 625 images was tagged by 49 subjects. During the data annotation session, an eye-gaze camera setup was used to track the eye-gaze of each subject when they were presented with a stream of images. The goal of this setup was to analyze how much eye gaze contributed to image memorability. The model achieved an accuracy of only 61.48% when the ground truth eye gaze and predicted eye gaze were compared.

A novel multiple instance-based deep CNN for image memorability prediction was proposed that shows the performance levels that are close to human performance on the LaMem dataset. The model shows EMNet, automatically learns various object semantics and visual emotions using multiple instance learning frameworks to properly understand the emotional cues that contribute extensively to the memorability score of an image (*Basavaraju & Sur, 2019*).

The main problem with the previously proposed state of the art models is that they are computationally intensive. Some of the previously proposed models are not suitable for production purposes. Most of the previously proposed models constitute several pre-processing stages and use multiple CNNs in a parallel manner to provide results. The issues that accompany these strategies are over-fitting, high computational complexity and high memory requirements. To solve these issues, there is a need for an approach that results in a smaller number of parameters and a model that contains layers that can prevent overfitting.

Therefore, to solve the above-mentioned issues, in this work, the proposed Residual Memory Net (ResMem-Net) is a novel deep learning architecture that contains fewer parameters than previous models, making it computationally less expensive and hence is also faster during both training and inference. ResMem-Net also uses 1x1 convolution layers and Global Average Pooling (GAP) layers, which also helps to reduce the chances of overfitting. In this model, a hybrid of Convolutional Neural Networks and Long Short-Term Memory Networks (LSTM) is used to build a deep neural network architecture that uses a memory-driven technique to predict the memorability of images. ResMem-Net achieves results that are very close to human performance on the LaMem dataset. Transfer learning is also taken advantage of during the training process, which has helped ResMem-Net to generalize better.

The publicly available LaMem dataset is used to train the model, consisting of 60,000 images, with each image being labeled with a memorability score. The proposed architecture has given close to human performance with a rank correlation of 0.679 on the LaMem dataset. Finally, heatmaps have been generated using Gradient Regression Activation Map (GradRAM) technique (*Selvaraju et al., 2017*), which is used the visualize and analyze the portions of the image that causes the image to be memorable. Even though this paper focuses on the results of the LaMem and Isola et al. dataset, the key contribution of this paper is the novel ResMem-Net Neural Network architecture which can be used for any other classification or regression task in which the intermediate features of the CNN might be useful.

In the Materials & Methods section, the proposed architecture, the novelty of the architecture, loss function, and the datasets used are explained in detail. The use of transfer learning, optimization function, evaluation metrics, loss function and weight update rule are also discussed in the Materials & Methods sections. In the Experiments and Results section, the experimental setup, hyperparameters used, training settings the results of the model are discussed. The results are compared in detail with existing works and a qualitative analysis done to understand memorability is also discussed. Finally, in the Conclusions section, the proposed work and results are summarized and then the potential future enhancements are discussed.

# MATERIALS & METHODS

This section deals with the proposed Neural Network architecture, the dataset used, and the evaluation of the proposed model's performance. Further, the results obtained from an extensive set of experiments are compared with previous state-of-the-art results. It shows the superiority of the proposed architecture; for every problem solved by deep learning, four core entities have to be defined before the results are obtained. They are the dataset, the neural network architecture, the loss function and the training procedure.

## Deep hybrid CNN for the prediction of memorability scores

This section provides a detailed explanation of the ResMem-Net. A visual depiction is given in Fig. 1. The figure shows that there are two distinct portions in the entire architecture. At the top of ResMem-Net, ResNet-50 (*He et al., 2015*) is used as the backbone, state of

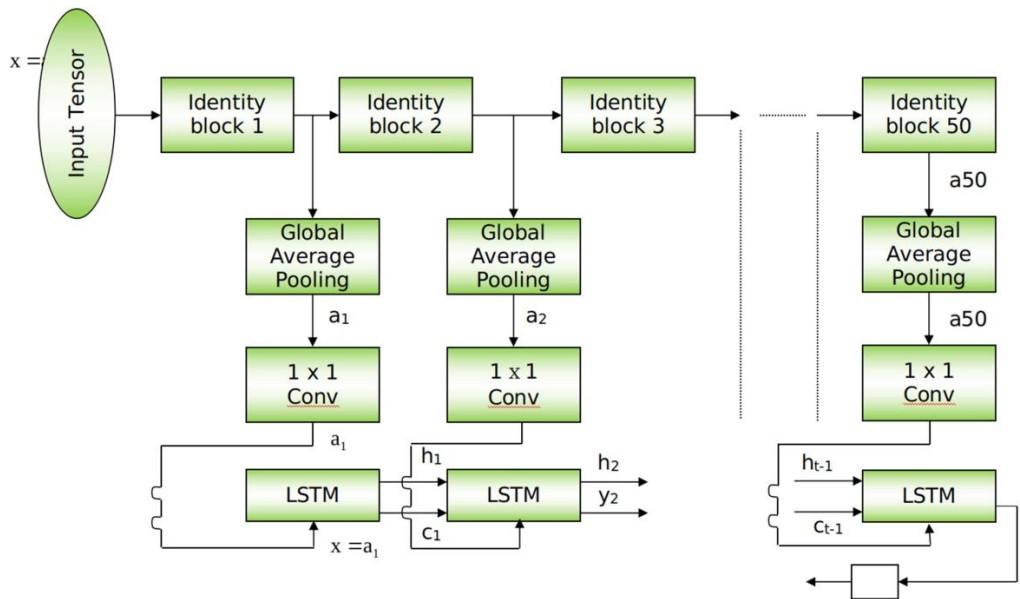

**Figure 1 Schematic for ResMem-Net.** Schematic diagram of the proposed architecture.

the art deep learning architecture for many applications. ResNet-50 is a 50-layer deep neural network that contains convolution kernels at each layer. The main innovation in ResNet-50 is the skip connection which helps to avoid vanishing gradients in very deep neural networks. The skip connection is present at every convolutional kernel present in the ResNet-50 model. The skip connection adds the input of the convolutional kernel to the output, hence allowing the model to propagate information to the next layer even if the output of the convolutional kernel is too small in terms of numerical value. This is how ResNet-50 and other variants of ResNet are not prone to vanishing gradient problem (*He et al., 2015*). Since there are going to be 50 convolutional kernels in ResNet-50, it is a Deep CNN. The input image is given to ResNet-50, and the size of the image used in our experiment is 224x224 px. One of the core features of the proposed architecture is that the CNN part of the architecture is fully convolutional, and due to the use of Adaptive Average Polling layers, the model isn't constrained to the size of the input image hence the input image can be higher or lesser than 224 × 224 px size.

At the bottom of ResMem-Net, a Long Short-Term Memory (LSTM) unit is responsible for predicting the output, the memorability score. LSTM is an enhancement to Recurrent Neural Networks (RNN). RNNs are generally used for sequential data such as text-based data or time-series data. However, in RNN, there are no memory units to resolve any long-term dependencies (*Cho et al., 2014*). Several variants of LSTM were analyzed, and it showed that the standard LSTM model with forget gate gave the best results on a wide variety of tasks (*Greff et al., 2016*). In an LSTM unit, a 'cell state' is computed that can retain information from previous input sequences. The cell state is computed using 'forget gate' and 'output gate' as demonstrated in Fig. 2. These gates determine which information

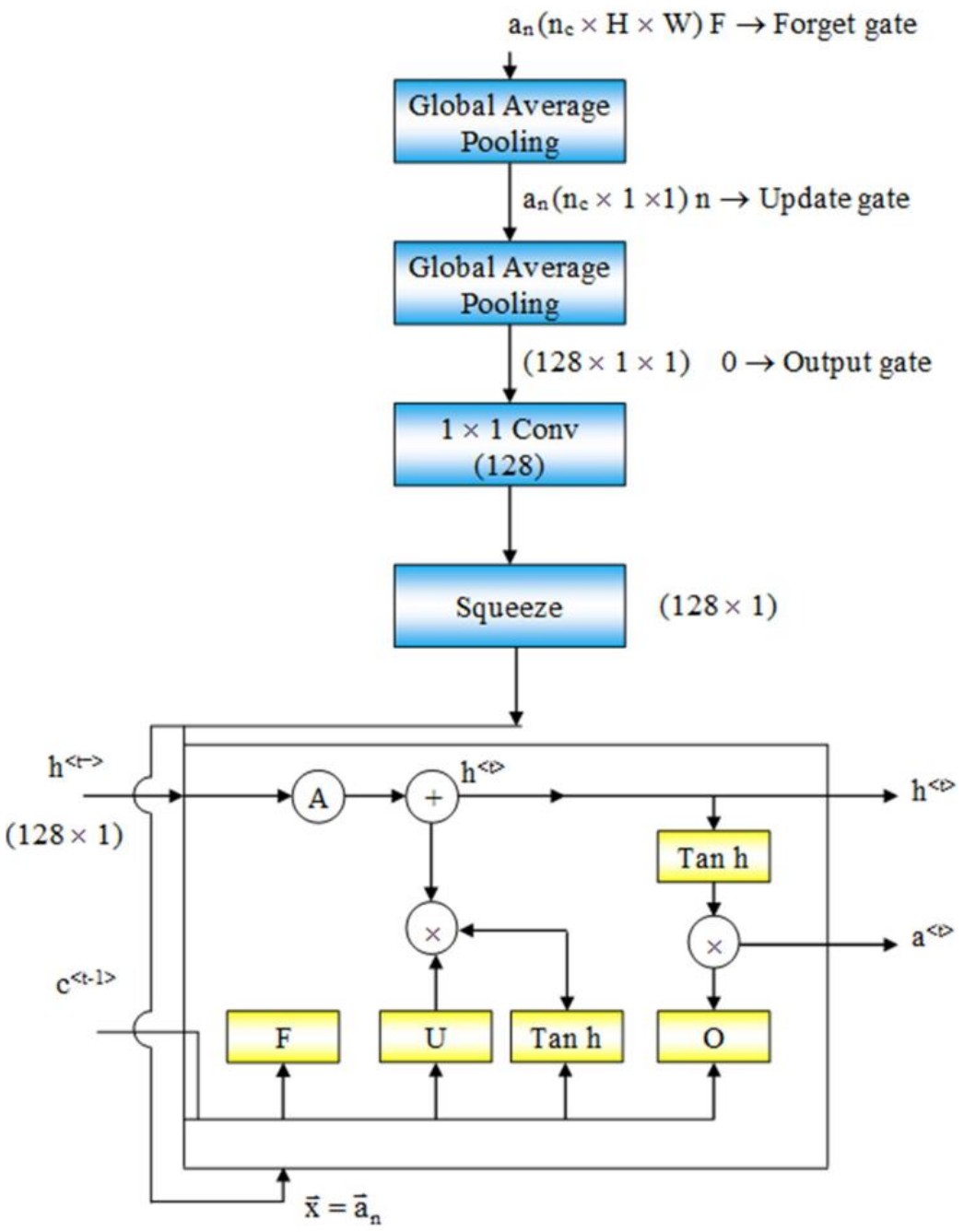

**Figure 2** **Flow of data from an intermediate layer of Resnet50 to the LSTM.** Flow chart of intermediate layer function.

from previous layers should be removed from the cell state vector and which information should be retained. LSTM units accept sequential data as inputs, and in this architecture, the input to the LSTM unit are the activations of the hidden layers of the ResNet-50 model, as shown in Fig. 2. As the input sequences being sent to the LSTM unit must be of the

same size, Global Average Pooling (GAP) is used to shrink the activations of the hidden layers to a size of $(C \times 1 \times 1)$ where C is the number of channels. Global Average Pooling is very much like Densely connected Layers in Neural Networks because it performs a linear transformation on a set of feature maps. This allows us to ensure that there is no need to care too much about the size of the output activations at each layer and the input image's size. As studied in *Hsiao et al. (2019)*, Global Average Pooling also does not have any parameter to optimize, thus avoiding overfitting and reducing computational needs. GAP layers can be thought of as an entity, that enforces the feature maps (outputs of intermediate layers) to be the confidence maps of various intrinsic features of the input image. Hence, GAP also acts as structural regularizers without requiring any hyper parameters. Also, global average pooling sums the spatial information, hence they are also robust to any spatial changes in the feature maps. Further, a convolution operation is done on the output of GAP layers to obtain a 128-channel output which can be flattened to obtain a vector of Rank 128.

The main reason behind passing the hidden layer activations to the LSTM unit is to ensure that the cell state vector can remember and retain the important information from the previous hidden layers. When the final layer's activation is passed to the LSTM unit, the important information of the previous layers along with the final layer's activation is obtained and then all that information is used to compute the memorability score. The LSTM layer's output is an n-dimensional vector, passed to a linear fully connected layer that gives a scalar output, which is the memorability score of the image. This strategy allows us not just to use the final layer's activations alone which is generally done in previous works discussed.

## Mathematical formulation of the model

So, the input image is a tensor of size (3, 224, 224), denoted by $A_0$. The output of Lth identity block is denoted by $A_L$, as shown in Eqs. (1) and (2). At each Lth identity block, the output of the identity block is calculated by:

$$Z_L = W_L \otimes A_{L-1} \tag{1}$$

$$A_L = relu(Z_L) \text{ where } relu(a) = max(0, a) \tag{2}$$

where,

$Z_L$ is the output of the Lth identity block,

$A_L$ is the output of the activation function with $Z_L$ as input.

For all L, $A_L$ is passed through a Global Activation Pooling layer, which converts a (C, W, H) tensor to a (C, 1, 1) tensor by taking the average of each channel in the activation matrix $A_L$.

At the LSTM layer, the initial cell state is denoted by $C_0$, and h0 denotes the initial activation. Before the hidden layer activations are passed to the LSTM, $C_0$ and $h_0$ are initialized as random vectors using 'He' initialization strategy to help avoid the exploding gradient problem (*He et al., 2015*).

The LSTM unit consists of three important gates that form the crux of the model:

1. Update Gate – Decides what information should be remembered and what information should be thrown away
2. Forget Gate – To decide which information is worth storing
3. Output Gate – The output of the LSTM unit

$$\text{Update Gate}: G_u = \text{sigmoid}(W_{hc}c^{<t-1>} + W_{ub}x^{<t>} + b_u) \tag{3}$$

$$\text{Forget Gate}: G_u = \text{sigmoid}(W_{fc}c^{<t-1>} + W_{fx}x^{<t>} + b_f) \tag{4}$$

$$\text{Output Gate}: G_o = \text{sigmoid}(W_{oc}C^{<t-1>} + W_{ox}x^{<t>} + b_u) \tag{5}$$

$$\text{Hidden cell state}: h^{<t>} = G_u + h^{<t>} + G_f * h^{<t-1>} \tag{6}$$

$$\text{LSTM output}: C^{<t>} = G_o * h^{<t>} \tag{7}$$

The output of $G_u$, $G_f$, $G_o$, $h^{<t>}$, and $c^{<t>}$ can be calculated using the formulas given in Eqs. (3), (4), (5), (6) and (7), respectively.

## The loss function

The scores of the images in both the mentioned datasets are continuous-valued outputs, making this entire task a regression task. To understand how good our model predicts memorability, loss functions are used, which can approximate the divergence between the target distribution and the predicted distribution. Generally, for regression tasks, the L2 loss function, also known as the Mean Squared Error (MSE), is used as the loss function for the proposed model and the formula is given in Eq. (8).

$$MSE = \frac{1}{n}\sum_{i=1}^{n}(y_i - \tilde{y}_i)^2 + \lambda\sum(\theta)^2 \tag{8}$$

where the $(\tilde{y})$ represents the predicted value, while $y_i$ represents the ground-truth value of the ith image in the dataset, $\lambda$ represents weight decay and $\theta$ represents the weights. The second term is added to the existing loss function to prevent the model from overfitting. The regularization procedure is known as L2 regularization, which multiplies a weight decay (hyperparameter) and the summation of all the weights used in the Neural Network. The weight decay prevents the weights from being too big, which ultimately prevents the model from overfitting.

## Pseudocode for ResMem-Net

The pseudocode for the forward-pass of ResMem-Net is given below. Initially, the information passed through each layer in the backbone by passing the previous layer's

output to the next layer. Each time an output from a layer is obtained, the outputs are passed to a global average pooling layer, which works as depicted in the function called globalAveragePooling. The outputs from the globalAveragePooling method are passed to the LSTM_CELL at each iteration. After the final iteration, the memorability score can be retrieved from the LSTM_CELL.

```
Procedure mem (images):
Cache = []
A[0] = images[0]
For i = 1 to n_layers:
    A[i] = W[i] (⊗) a[i-1] + b[i]
    A[i] = relu(A[i])
    A[i] = A[i] + A[i-1]
    Cache[i] = A[i]
For i = 1 to n_layers:
    S = cache[i]
    S = globalAveragePooling(S)
    S = W[i] (⊗) S
    h, c = LSTM_CELL(s, h, c)
L = w_l * h + b_l
Return L
```

```
Procedure LSTM_CELL(x, h_{t−1}, c_{t−1}):
it = sigmoid (W_{xi} * x + W_{hi} * h_{t−1} + W_{ci} * c_{t−1} + b_i)
ft = sigmoid (W_{xf} * x + W_{hf} * h_{t−1} + W_{cf} * c_{t−1} + b_f)
ct = ft* c_{t−1} + it * tanh(W_{hc} * h_{t−1} + W_{xc} * x + b_c)
ot = sigmoid(W_{xo} * x + W_{ho} * h_{t−1} + W_{co} * ct + b_o)
ht = ot * tanh(ct)
return ht, ct
```

```
Procedure globalAveragePooling(tensor):
c, h, w = dimensions(tensor)
for i in range(c):
    Avg = (1/h) * (Σtensor[i])
    tensor[i] = Avg
return tensor
```

In Fig. 3, the pipeline used during this research is depicted. The process starts with data collection and processing and then proceeds with the model development phase. In the model development phase, the model's architecture is initially defined, modified to our task and finally programmed. Then the training phase is done with the given datasets, and finally, hyper-parameter tuning is done, where various batch sizes, learning rates and residual models are tried to find the optimal settings. Then to analyze the results, GradRAM technique is used to visualize the activation maps to understand how the model predicts the results.

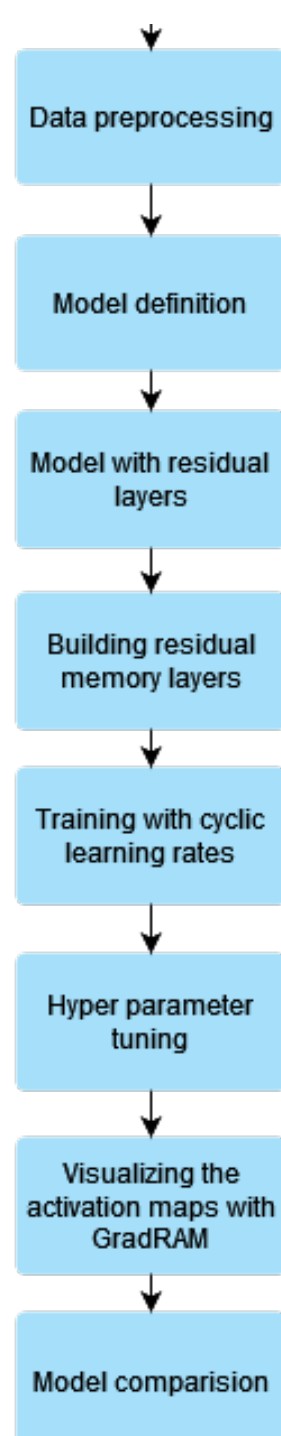

**Figure 3** **Pipeline used in this work.** Pipeline illustration.

## Dataset used

In this paper, two publicly available datasets are used: the LaMem dataset and dataset from Isola et al. LaMem is currently the largest publicly image memorability dataset that contains 60,000 annotated images. Images were taken from MIR Flickr, AVA dataset, Affective images dataset, MIT 1003 dataset, SUN dataset, image popularity dataset and Pascal dataset. The dataset is very diverse as it includes both object-centric and scene-centric images that capture a wide variety of emotions. The dataset from Isola et al. contains 2,222 images from the SUN dataset. Both datasets were annotated using the Visual Memorability Game. Amazon Mechanical Turk was used to allow users to view the images and play the game which helped annotate the images.

Both the datasets were collected with human consistency in mind, *i.e.,* the authors ran human consistency tests to understand how consistent the users are able to detect repetition of images. The consistency was measured using Spearman's rank correlation, and the rank correlation for LaMem and Isola et al. are 0.68 and 0.75 respectively. The human consistency was calculated by inviting a new set of participants to play the Visual Memorability Game. The participants were split into two halves and were asked to independently play the game for the images in the datasets. Then, the human consistency was measured by how similar the second half the participants' memorability scores were to the memorability scores obtained from the first half of the participants. This analysis show that humans are generally consistent when it comes to remembering or forgetting images. Also, for both the datasets, the authors of the datasets have themselves provided the dataset splits along with the dataset. Those files contain both the ground truth values of each image and information about whether they belong to the training or validation sets. In the LaMem dataset, 45,000 images are given for training, while 10,000 images for validation.

## Optimization

The loss function is actually differentiable and is also a function of the parameters of the Neural Network. The gradient of the loss function concerning the weights can guide us through a path to allow us to identify the right set of parameters that yield a low loss using gradient descent-based methods. In our experiments, a slightly modified method of ADAM optimizer is used, which is a combination of Stochastic Gradient Descent with Momentum and RMSprop added with the cost function (*Yi, Ahn & Ji, 2020*) The loss function of Neural Networks is very uneven and sloppy due to the presence of too many local minima and saddle points. This modified version of ADAM uses exponentially weighted moving averages. Initially, compute the momentum values are computed using Eqs. (9), (10) and (11):

$$H(\theta) = (\theta) + \frac{\partial J(\theta)}{\partial \theta} \tag{9}$$

where, J - cost function

$\lambda$ - scaling constant (hyperparameter)

$\theta$ - weights

$$m_i = \alpha m_{i-1} + (1 - \alpha)H(\theta_i) \tag{10}$$

$$v_i = \beta v_{i-1} + (1-\beta)(H(\theta i))^2 \tag{11}$$

where, $\alpha$, $\beta$ - scaling constant
$m_i$, $v_i$ –first, second momentum
$m_0$, $v_0$ –initial momentum (set to 0)
After the momentum values are calculated, the update rule for the weights is done using Eq. (12),

$$\theta_{i+1} = \theta_i - \eta \frac{\hat{m}_i}{\sqrt{\hat{v}_i + \varepsilon}} \tag{12}$$

Where, $\theta_i$ - current weight
$\theta_{i+1}$ - updated weight
$\eta$ - learning rate

$$\hat{m}_i = \frac{m_i}{(1 = \alpha)}; \; \hat{v}_i = \frac{v_i}{(1 - \beta)}$$

$\varepsilon$ - constant to avoid zero division (usually $10^{-6}$)
Adding the cost function to the gradient of the weights w.r.t the cost functions ensures that the loss landscape is much smoother and can converge at a good minimum. This helps because, even if the gradient of the cost function w.r.t weights are very small, adding the scaled version of the cost function ensures that the weights keep changing, ensuring that the model doesn't get stuck in local minima or saddle points.

## Learning rate and one cycle learning policy

The learning rate is one of the most important hyperparameters in deep learning as it decides how quickly the loss moves towards a minimum in the loss function's surface. Learning rates can decide whether a model converges or diverges over time. If a high learning rate is used throughout the training process, loss of the model may diverge over some time, but if it is set to a low value, then the model may take too much time to converge. To solve this issue, generally the learning rate is reduced over time using a decaying function. But decaying functions can lead to the model's parameters to be stuck in saddle points or in local minima, which can lead to the model not learning new parameters in the consecutive epochs. To avoid these issues, (*Smith, 2018*) has proposed a method called One Cycle Learning. In one cycle learning policy, for each epoch, the learning rate is varied between a lower bound and upper bound. The lower bound's value is usually set at 1/5th or 1/10th of the upper bound.

In one cycle learning, each epoch is split into 2 steps of equal length. All deep learning models are trained using mini-batches, so if the dataset has 100 batches, the first 50 batches are included in step 1 and the rest are included in step 2. During the start of each epoch, the learning rate is set to the lower bound's value and at the end of each mini-batch, the learning rate is slowly increased to ensure that the learning rate reaches the upper bound by the end of step 1. In step 2, the training proceeds with the upper bound as the learning

rate and then the learning rate is slowly decayed after each mini-batch, to ensure that by the end of step 2, the learning rate is back to lower bound. This is then repeated for each epoch. Varying the learning rate between a high and a low value allows the model to escape the local minima or saddle points. The higher learning rate allows the model to escape local minima and saddle points during training, while the lower learning rate ensures that the training leads to parameters that ensure a lower loss in the loss function.

### Transfer learning

Transfer learning is training a model on a large dataset and retraining the same model on a different dataset with lesser data. Intuitively, the learned features from larger datasets are used to help improve accuracy on datasets with smaller data points. In our work, a pre-trained ResNet-50 that is trained on the ImageNet dataset is used, which contains 3.2 million images, with each image categorized in one among the 1000 categories.

In our work, the semantic features learned through ImageNet will allow the model to be quickly trained and perform better on identifying the memorability of images in the validation dataset. The feature maps in the pretrained ResNet-50 will contain feature maps for objects, scenes and other visual cues that aren't present in the images present in the datasets that are used to train the model. This is so because, the pretrained ResNet-50 was trained on a dataset with diverse set of images. So, after careful retraining, many of these feature maps in the pretrained model will be retained. This will allow the re-trained model to identify the objects and scenes not present in the LaMem and Isola et al. datasets, which can drastically improve real-time deployment performance. Empirical evidence for the above explanation is given in *Rusu et al. (2016)*.

## EXPERIMENTS AND RESULTS

In this section, the evaluation criteria, and outcome of the experiments are discussed. The training settings i.e, the hyperparameters, hardware used, training and validation splits of the dataset are discussed. Finally, the outcome of the training process and the reason behind superior results are explained.

### Evaluation metric

The L2 loss function is generally a good metric to find how well the proposed model performs, but here the Rank Correlation method is also used to evaluate the proposed model. The Spearman Rank Correlation ($\rho$) is computed between the predicted score and target score, is used to find the consistency between the predicted scores and target score from the dataset. The value of $\rho$ ranges from $-1$ to 1. If the rank correlation is extremely close to 1 or $-1$, then it means that there is a strong positive or negative agreement respectively between the predicted value and ground truth, while a rank correlation of 0 represents that there is complete disagreement. The rank correlation between predicted and target memorability score is given by the Eq. (13):

$$\rho = 1 - \frac{6 \cdot \sum_{i=1}^{n}(r_i - S_i)^2}{n^3 - n} \tag{13}$$

**Table 1  Performance of various architectures on Lamem dataset.** ρ values compared.

| Method (LaMem dataset) | (ρ) |
| --- | --- |
| ResMem-Net (proposed model) | 0.679 |
| MemNet | 0.640 |
| MCDRNet | 0.663 |
| EMNet | 0.671 |
| CNN-MTLES (different dataset split) | 0.5025 |

where $r_i$ is the ground truth while $s_i$ is the predicted value from the model, whereas $n$ is the number of images in the dataset.

## Training settings and results

The batch size was set at 24 throughout the training process and the images were resized to a size of $224 \times 224$. Since transfer learning is employed, when the backbone's (ResNet-50) parameters were frozen, the upper bound and lower bound for the learning rate was set at 0.01 and 0.001 respectively. After 10 epochs, the backbone's parameters were unfrozen and then the upper bound and lower bound for the learning rate was set at 0.001 and 0.0001 for 15 epochs. For the rest of the epochs, the lower bound and upper bound for the learning rate were set at 0.0001 and 0.00001 respectively. The training process consisted of a total of 40 epochs. For regularization, a value of 0.0001 was set as L2 weight decay. The model was trained on a Nvidia Quadro P5000 GPU which has 16GB GPU memory and 2560 CUDA cores.

To ensure stable training, normalization and dropout layers were used. The authors of the LaMem dataset have given 5 training set splits because each image has multiple annotations. Hence, 5 different models were trained, one for each split, and the results were averaged across the models while testing. For cross-validation purposes, the authors of the LaMem datasets divided the dataset into five sets, where each set contains 45,000 images for training, 10,000 images for testing and 3,741 images for validation purposes. After the training the model using the above settings, ResMem-Net obtained an average rank correlation of 0.679 on the LaMem dataset and 0.673 on the Isola et al. dataset as mentioned in Tables 1 and 2. These results indicate that the use of a hybrid of pretrained CNN and LSTM has contributed to the increase in the accuracy of the model. Also, the model contains only a ResNet-50 backbone and a LSTM unit, making it computationally less expensive.

## DISCUSSION

In this section, the results of the experimental outcomes are taken and compared with existing models on two datasets, namely, LaMem and the Isola et al. dataset. Then, the comparison the number of parameters present in the existing models and the proposed model is done to establish why the proposed model has lesser memory requirements and to establish why the model is better suited for deployments to servers or other production needs. Finally, the results of the qualitative analysis done using the GradRAM method is

**Table 2** **Performance of various architectures on Isola et al dataset.** ρ values compared with another dataset.

| Method (Isola et al. dataset) | (ρ) |
| --- | --- |
| ResMem-Net (proposed model) | 0.673 |
| MemNet | 0.61 |
| MCDRNet | 0.638 |
| EMNet | 0.664 |
| SVR | 0.462 |

presented to understand which regions of the image lead to higher memorability scores and to answer the question, "What makes an image more memorable?".

The Spearman Rank Correlation metric has been used to evaluate the models and consistency of the results. Since each image has been annotated by multiple subjects, the rank correlation metric is better suited than the L2 loss to compare how consistently the models are predicting memorability scores. The five models discussed in the introduction are considered and the average of results is compared with the previous works. To ensure that the comparison of our results is fair, as mentioned in results sections, five models were trained on the five sets and found the average of the results. The models used for comparison were also trained in the same way by the respective authors, hence that ensures that the differences between the results of the previous models is due to the models only, not due to any other reasons.

Table 1 represents the results of various models on the LaMem dataset and Table 2 represents results of various models on the Isola et al. dataset. Table 1 and Fig. 4 show that ResMem-Net attains a rank correlation of 0.679, a 6.09% increase from MemNet, a 35% increase from CNN-MTLES, a 2% increase from MCDRNet and a 1.2% increase from EMNet. The human-level accuracy on LaMem is 0.68, and ResMem-Net has brought us extremely close to human accuracy with a difference in rank correlation of just 0.001.

From Table 2 and Fig. 5, it can be inferred that ResMem-Net attains a rank correlation of 0.673, which is a 10.33% higher from MemNet, 5.48% increase from MCDRNet, 1.4% increase from EMNet and a 45.67% increase from SVR. The authors have not provided the human accuracy for this dataset. Hence, it is not possible to tell how close ResMem-Net is to human accuracy for the Isola et al. dataset, but it is clear that ResMem-Net has outperformed all other previous works. The reason behind the superior performance can be attributed to the use of LSTM unit, modified optimization function, pretrained ResNet-50 backbone and the use of cyclic learning rates.

Tables 3 and 4 depict the predicted scores on various sets of images on the dataset. In both the tables, the images are arranged in descending order of the predicted memorability scores. For example, the 'Top 10' row depicts the average of the top 10 highest predicted memorability scores that are predicted by various network architectures and finally, the average of the ground truth of the same images is also given in the same row. The results are based on average over the 5-fold cross-validation tests as provided by the creators of the datasets.

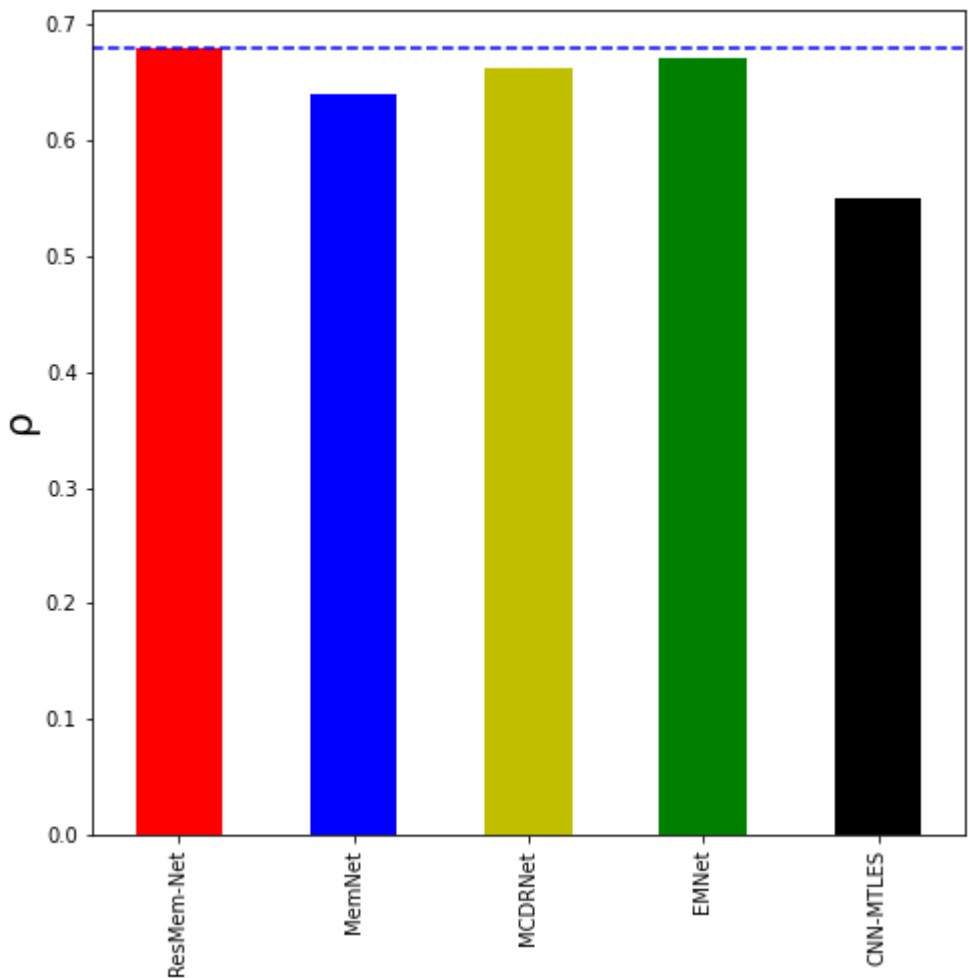

**Figure 4** **Bar chart showing the superiority of ResMem-Net over previous works on the LaMem dataset.** The results shown are the average of the rank correlation obtained across the validation splits. Bar chart comparison with first dataset.

From both Tables 3 and 4, it can be inferred that, on average, ResMem-Net performs better than previously proposed models on both Isola et al. dataset and LaMem dataset. For the top 10, EMNet predicts a memorability of 91.89% and 82.43% on the LaMem and Isola et al. datasets, respectively, while ResMem-Net predicts an average memorability of 93.82% and 82.61% on the same LaMem and Isola et al. datasets, respectively. MCDR-Net obtained an average memorability 93.15% and 81.75%, while MemNet has obtained 91.7% and 80.16% on the LaMem and Isola et al. datasets respectively for the top 10. When compared to the ground truth, which is 100%, these scores clearly state that ResMem-Net is more consistent with the images with high memorability. On the other hand, when for the 'Bottom 10' images, EMNet predicts an average memorability of 48.41% and 27.42%, while for MCDRNet it is 50.94% and 26.52% on the LaMem and Isola et al. datasets, respectively. In comparison, ResMem-Net predicts an average memorability of 47.9% and 27.42%, respectively. Again, when comparing those results to the ground truth values,

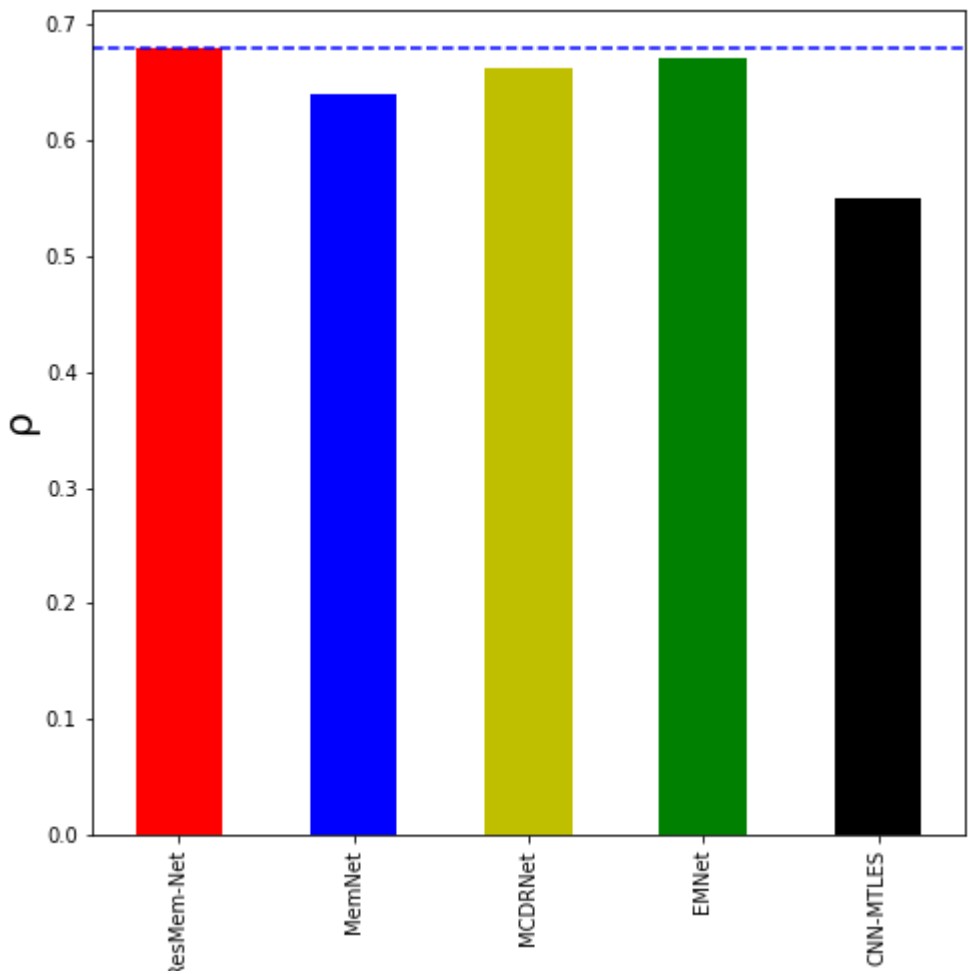

**Figure 5** **Bar chart showing the superiority of ResMem-Net over previous works on the Isola et al. dataset.** The results shown are the average of the rank correlation obtained across the validation splits. Bar chart comparison with another dataset.

which is 33.57% and 5.69% for LaMem and Isola et al. datasets respectively, ResMem-Net provides similar results to EMNet.

It would also be unfair to completely ignore the results of traditional machine learning algorithms on image memorability. Despite many empirical results that depict the superiority of deep learning algorithms on computer vision tasks, certain studies have shown that the use of hand-crafted features when ensembled with machine learning algorithms such as SVR or Random Forests can, in fact, provide better results. Of course, concerns regarding the generalization of the models on new data have been raised, which are the very papers that propose the non-deep learning-based strategies themselves. However, in Table 5 and Fig. 6, it is very clear that the proposed ResMem-Net quite easily outperforms traditional machine learning strategies.

Since the results are from the validation set, it is clear that the model did not overfit but rather learned features that contribute to the memorability scores of the image. The

**Table 3 Comparison of predicted scores and ground truth scores for various sets of images from the LaMem dataset.** Comparison table between different networks with the proposed one.

| Dataset | MemNet | MCDRNet | EMNet | ResMem-Net | Ground Truth |
|---|---|---|---|---|---|
| **Top 10** | 91.70% | 93.15% | 91.89% | 93.82% | 100% |
| **Top 25** | 90.40% | 91.76% | 91.36% | 92.8% | 100% |
| **Top 50** | 89.57% | 91.27% | 90.85% | 90.52% | 99.35% |
| **Top 100** | 89.17% | 90.42% | 90.38% | 90.74% | 98.45% |
| **Top 200** | 88.91% | 90.00% | 90.02% | 90.65% | 97.57% |
| **Bottom 200** | 55.06% | 54.57% | 54.23% | 52.18% | 42.16% |
| **Bottom 100** | 54.35% | 52.79% | 52.74% | 49.13% | 39.01% |
| **Bottom 50** | 54.20% | 51.65% | 51.25% | 48.72% | 36.3% |
| **Bottom 25** | 54.44% | 51.20% | 50.34% | 48.27% | 34.41% |
| **Bottom 10** | 58.06% | 50.94% | 48.41% | 47.9% | 33.57% |

**Table 4 Comparison of predicted scores and ground truth scores for various sets of images from the Isola et al. Dataset.** Comparison of different networks with the proposed one with another dataset.

| Dataset | MemNet | MCDRNet | EMNet | ResMem-Net | Ground Truth |
|---|---|---|---|---|---|
| **Top 10** | 80.16% | 81.75% | 82.43% | 82.61% | 96.54% |
| **Top 25** | 75.46% | 79.77% | 81.41% | 82.14% | 94.39% |
| **Top 50** | 75.13% | 78.57% | 79.48% | 80.27% | 92.24% |
| **Top 100** | 74.32% | 76.64% | 77.63% | 79.76% | 89.59% |
| **Top 200** | 73.58% | 74.9% | 76.64% | 77.92% | 85.33% |
| **Bottom 200** | 35.91% | 34.83% | 34.08% | 35.95% | 22.85% |
| **Bottom 100** | 32.8% | 31.64% | 31.66% | 31.4% | 18.66% |
| **Bottom 50** | 30.14% | 29.41% | 28.94% | 29.67% | 14.93% |
| **Bottom 25** | 28.81% | 26.6% | 29.47% | 29.23% | 10.95% |
| **Bottom 10** | 28.29% | 26.52% | 26.86% | 27.42% | 5.69% |

**Table 5 Comparison of accuracy of ResMem-Net and other traditional machine learning methods on the Isola et al. dataset.** Comparison with traditional machine learning algorithms.

| Method | $(\rho)$ |
|---|---|
| Isola et al. | 0.46 |
| Khosla et al. (b) (*Khosla et al., 2012*) | 0.50 |
| Mancas and Meur | 0.479 |
| Peng et al. (*Peng et al., 2015*) | 0.52 |
| ResMem-Net | 0.673 |

validation set also encompasses a wide variety of landscapes and events, which also leads us to believe that the model performs well on different kinds of images.

## Computational complexity analysis

This section deals with the comparison of the computational complexity of various previous models and ResMem-Net. It has been already established that ResMem-Net is quite minimal compared to other previously proposed models in network parameter size.

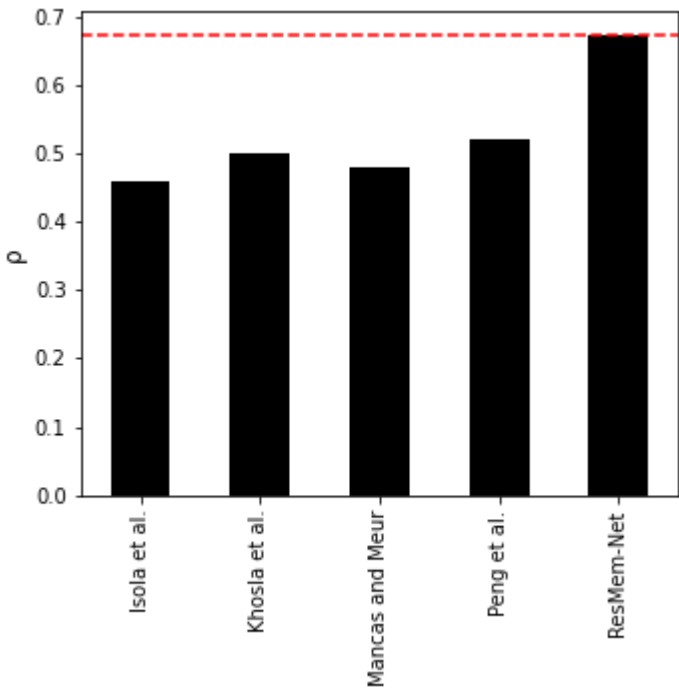

**Figure 6** **Bar chart showing the accuracy of non-deep learning based models.** The rank correlation values shown are the average of the rank correlation obtained across the validation splits.

**Table 6** **Computational complexity analysis of proposed and existing models.** Computation complexities comparison.

| Model | No. of parameters |
|---|---|
| ResMem-Net | 26 million ≈. |
| MemNet | 60 million ≈. |
| MCDRNet | 276 million ≈. |
| EMNet | 414 million ≈. |

Table 6 shows the number of parameters or weights present in ResMem-Net, MemNet, MCDRNet and EMNet. It is clear from the table that ResMem-Net has a significantly much lesser number of parameters than the previously proposed network architectures. CNN's are composed of convolution operations, which are very much compute intensive. So, the lesser the number of parameters, the faster the model takes to provide outputs. The bigger advantage of ResMem-Net is that it has a significantly lesser number of weights and still provides better accuracy on both LaMem and Isola et al. datasets. The time taken for ResMem-Net to process an image of size $512 \times 512$px is approximately 0.024s on Nvidia Quadro P5000 GPU. It should also be noted that having too many parameters can cause overfitting and hence ResMem-Net is less prone to overfitting because it has a significantly lower number of weight parameters.

## Qualitative analysis of the results

In this section, the inferences and patterns that were identified after visually analyzing the results of ResMem-Net are discussed. To aid us with this process, GradRAM technique was used to understand which part of the images are focused by ResMem-Net or in other words, which part of the image gives larger activations. GradRAM is an extension of Class Activation Maps (CAM), which uses the gradient obtained during backpropagation process, to generate heatmaps. The heatmaps generated shed light on which part of the image enhances the image's memorability. This shows how the hidden layers in the ResNet-50 backbone outputs feature maps and it is this information that has is being used by the LSTM unit to make predictions. Based on the heatmaps and careful manual analysis of the results using randomly selected images for different categories from the Isola et al. dataset, the following inferences are made:

The object in the image contributes more to the memorability score than the scene in which the object is placed. In almost every heatmap, it is observable that the portion of the image containing the main object provides higher activations compared to the rest of the image. Also, images with no objects are predicted to be less memorable compared to images containing objects (both living and non-living). Also, images containing a single central object is seen to be more memorable than images with multiple objects. The average rank correlation of the predicted memorability of the images with a single central object is 0.69, while the average of the rank correlation of the images without a central object is 0.36. Also, the presence of humans in the image contributes to a better memorability score. If the human in the image is clearly visible, then the memorability averages at 0.68, while if the image does not contain any human or object, then the memorability averages at 0.31.

Using a model pretrained on object classification datasets provide better results and trains faster than using a model pretrained on scene classification datasets (*Jing et al., 2016*). This can be attributed to the fact that memorability scores are directly related to the presence of objects. Thus, a model whose weights contain information about objects take lesser time to converge to a minimum (*Best, Ott & Linstead, 2020*). Also, image aesthetics does not have much to do with Image memorability. A few images containing content related to violence are not aesthetically good, but the memorability score of the image is high, with an average memorability of 0.61.

## CONCLUSIONS

Capturing memorable pictures is bit challenging as it requires an enormous amount of creativity. However, just like any other phenomenon in nature, humans' capability to remember certain images more follows a pattern. This paper becomes unique by introducing ResMem-Net, a novel neural network architecture that combines a pretrained deep learning model (ResNet-50) and a LSTM unit. The model was trained using the One Cycle Learning Policy, which allows the use of cyclic learning rates during training. ResMem-Net has provided a close to human performance on predicting the memorability of an image using the LaMem dataset, which is the largest publicly available dataset for image memorability. The rank correlation of ResMem-Net is 0.679, which is extremely

close to human accuracy 0.68. This obtained result 6.09% increase the performance of MemNet, 35% increase from CNN-MTLES, 2% increase from MCDRNet and a 1.2% increase from EMNet. Based on the qualitative analysis executed using GradRAM method, it was inferred that the object plays a bigger role in enhancing the memorability of the image. A pre-trained model that consists of weights from an object classification dataset converges quickly than a model pre-trained on scene classification. These results were observed manually by looking through the highly rated images and lowly rated images. Heatmaps generated using the GradRAM method was also used to analyze and obtain the above inferences.

The limitation of the current work is that even though the model contains much lesser number of parameters than other state-of-the-art models, ResMem-Net is still not deployable to mobile based GPUs. To solve this issue, further research can be done to use mobile compute efficient architectures like MobileNetV3 or EfficientNet, which are also pre-trained on the ImageNet dataset. Further research can also be done to improve the accuracy of the model by replacing ResNet-50 with more recent architectures like ResNext. The LSTM unit can also be replaced with more recent architectures like the Transformer architecture or BiDirectional RNNs. A more generic suggestion is to spend time to develop larger datasets for image memorability prediction because with larger datasets, neural networks can generalize better.

### Funding
The Deanship of Scientific Research at Umm Al-Qura University supported this work by Grant Code: 19- ENG-1-01-0015. The funders had no role in study design, data collection and analysis, decision to publish, or preparation of the manuscript.

### Grant Disclosures
The following grant information was disclosed by the authors:
Umm Al-Qura University: 19- ENG-1-01-0015.

### Competing Interests
Ali Kashif Bashir is an Academic Editor for PeerJ.

### Author Contributions
- Arockia Praveen conceived and designed the experiments, performed the experiments, performed the computation work, prepared figures and/or tables, and approved the final draft.
- Abdulfattah Noorwali conceived and designed the experiments, prepared figures and/or tables, proof Reading, and approved the final draft.
- Duraimurugan Samiayya conceived and designed the experiments, authored or reviewed drafts of the paper, and approved the final draft.
- Mohammad Zubair Khan analyzed the data, prepared figures and/or tables, proof Reading, and approved the final draft.

- Durai Raj Vincent P M performed the experiments, authored or reviewed drafts of the paper, and approved the final draft.
- Ali Kashif Bashir analyzed the data, authored or reviewed drafts of the paper, and approved the final draft.
- Vinoth Alagupandi performed the computation work, prepared figures and/or tables, and approved the final draft.

## Data Availability

The data is available at http://memorability.csail.mit.edu/download.html

The Isola et al. dataset is available at: https://web.mit.edu/phillipi/Public/WhatMakesAnImageMemorable/.

The code is available at GitHub: https://github.com/praveenbenedict/ResMemNet.

## Supplemental Information

Supplemental information for this article can be found online at http://dx.doi.org/10.7717/peerj-cs.767#supplemental-information.

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
