# Peer review of "ResMem-Net: memory based deep CNN for image memorability estimation"

_PeerJ Computer Science, doi:10.7717/peerj-cs.767_

## Round 0.1 · original submission · Major Revisions

The authors need to incorporate all the comments given by the reviewers in the revised version.

Reviewers 1, 2 and 3 have requested that you cite specific references. I must direct you to NOT add these citations.

Reviewer 1 ·

Basic reporting

.

Experimental design

.

Validity of the findings

.

Additional comments

• This paper discusses the image memorability using Deep CNN and the work presented in this paper reads well and is logically structured in a clear narrative. The paper has a descriptive title, abstract, and keywords. Still, I have the following comments and observations that need to be accommodated or addressed in this paper:
• It is better to state the main motivations and contributions of this paper more clearly in the introduction. The novelty of the proposed work should be better emphasized.
• In One Cycle Learning, the use of the words Cycle, iterations, and epoch seems ambiguous and not properly explained.
• It is also not clearly mentioned in what way One Cycle Learning leads to better model fitting.
• In addition, the authors should include the details about the hyper-parameter settings.
• The evaluation metrics mention that the rank correlation of -1 represents complete disagreement, which I don't think is the case. The statement can be rechecked and rewrite accordingly.
• Regarding the image classification and segmentation, more literature needed to give a provide stronger backing to your claims. Check the below work!!
CL Chowdhary, et al., Analytical study of hybrid techniques for image encryption and decryption, Sensors 20 (18), 5162, 2020.
J Tamang, et al. Dynamical properties of ion-acoustic waves in space plasma and its application to image encryption - IEEE Access, 2021

Reviewer 2 ·

Basic reporting

no comment

Experimental design

no comment

Validity of the findings

no comment

Additional comments

1. The English language in the paper has to be polished.
2. What are the limitations of the existing works that motivated the current work?
3. Discuss the drawbacks of the current work in the conclusion.
4. There must be maximum two paragraphs in conclusion section. The first paragraph is for briefly discussing the entire paper and the second paragraph is for discussing some future works.
5. Short form must be defined as properly when used for the first time.
6. Never use I, we, you, our, etc. in a research article. Use one “Tense” to write the entire paper. “Present Tense” is preferable.

7. All the figures, tables, equations and references must be cited in the text.
8. References must be cited sequentially starting from 1, then 2, then 3 and so on.
9. The discussion is very important in the research paper. Nevertheless, this section is short and should be presented completely.
10. Please cite below references in introduction and related work.

“Median filtering forensics scheme for color images based on quaternion magnitude-phase cnn
“CNN approaches for classification of Indian leaf species using smartphones
An efficient image analysis framework for the classification of glioma brain images using CNN approach,
Early detection of diabetic retinopathy using PCA-firefly based deep learning model

Reviewer 3 ·

Basic reporting

- Paper is well written. Authors should add a little background of the study and limitations of the existing works and clearly explain the contributions at the end of the introduction.
- Spell out each acronym the first time used in the Abstract as well as the body of the paper.
- The objectives of this paper need to be polished. Contribution list should be polished at the end of the introduction section and last paragraph of the introduction should be the organization of the paper.

Experimental design

- The authors should clearly mention all parameters used to evaluate the performance of the model.
- The procedures and analysis of the data are seen to be unclear.
- Authors should clearly describe splitting criteria.
_ Should discuss more about the dataset.

Validity of the findings

- Major contribution was not clearly mentioned in the conclusion part.
- The discussion is very important in research paper. Nevertheless, this section is short and should be presented completely.

Additional comments

- Please improve the overall readability of the paper.
- Authors should add the most recent reference:
1) Byte-level object identification for forensic investigation of digital images, 2020 International Conference on Cyber Warfare and Security (ICCWS), 1-4
2) Cross corpus multi-lingual speech emotion recognition using ensemble learning, Complex & Intelligent Systems, 1-10
3) CANintelliIDS: Detecting In-Vehicle Intrusion Attacks on a Controller Area Network using CNN and Attention-based GRU, IEEE Transactions on Network Science and Engineering

·

Basic reporting

The article looks good. The detailed comments are mentioned in the attached file.

Experimental design

Some improvements are required, are mentioned in the attached file.

Validity of the findings

The experiments performed are valid, some more comparisons are required to show it more better.

Additional comments

All comments are mentioned in the attached file.

Reviewer 5 ·

Basic reporting

This paper addresses the interesting question of what makes an image memorable and uses state-of-the-art computational models in its quest to better understand why certain images are remembered better than others. The paper introduces a novel neural network architecture which can be used to better approximate, relative to other models, the memorability of images. I enjoyed this paper and think it has the ability to contribute a useful model to the literature. In addition to proposing a novel model, the authors utilize a clean approach consisting of publicly available datasets to provide support for the utility of their neural network design and its applicability to other researchers. I do, however, have a few concerns about the paper in its current form. Below I have listed these specific concerns, as well as suggestions as to how, in my opinion, the paper could be improved. I have organized these concerns relating to both theoretical considerations and content, as well as writing and style.

1) I would recommend revising your writing with respect to memorability, so that it is a bit cleaner and easier for the reader to follow. In lines 91-92 you describe “measure an individual's probability of remembering an image”, but then later you discuss that memorability is a property of the image. Try to be clearer about whether the memorability you are focused on in this work is a property of the image or of a person. The phrasing in line 95 implies that memorability could be tied to the person, rather than the image.

2) In line 200: “The main innovation in ResNet”, is this your innovation or the work of the creators of ResNet? If so, please cite.

3) Pertaining to lines 115-116, in the previous paragraphs you discussed how memorability could vary between people, provide a bit more detail about how this memorability score here was annotated.

4) In line 174, what is the “human performance” you are referring to here? Can you describe where this comes from? And then later in line 410, is this the same human accuracy referenced here? If not, please describe.

5) Pertaining to lines 373-375, can you explain more about how real-time deployment performance can drastically improve for objects and scenes that are not in the new datasets?

6) In line 107, what type of deployment do you mean here? Are you referring to the application of it?

Please include citations to support the various claims made in lines:
⁃ Line 89-90
⁃ Line 104-109
⁃ Line 134 (and remove the phrase “which was already mentioned“)
⁃ Line 487 - 493

A few instances of repeated words or phrasing
⁃ Line 399-400 “cross-validation purposes”
⁃ Line 497-499 “humans’ capability…”

Re-word/Re-phrase to make comprehension easier for the reader
⁃ Line 41 “a hardest problem”
⁃ Line 89 & 334 “prove”
⁃ Line 110-111 rephrase “was given”
⁃ Line 122-123 “greedy algorithm” -- I am not sure what you mean by this
⁃ Line 159-160 and 165-166 both statements “propose” the new model. Can you remove one of these to make this clearer?

Experimental design

The methods and approach you described are technically sound and fits with the general scope of this journal.

1) In line 45 in the Abstract, you note the “information from the hidden layers” — could you speak to what type of information is stored in each layer? Can you connect this to the qualitative section later in the paper where you discuss contributions to memorability such as the presence of an object within the scene and the presence of a human, etc.
2) Can you provide data and/or a quantifiable measure to support the claims in lines 481-485. For example, instead of stating that manually you observed a difference with respect to the presence of a human or not, you could quantify the average memorability for images containing a human vs without a human.
3) In line 436, please report the ground-truth values here like you did in line 432 for the Top 10

Validity of the findings

The findings and data reported are appropriate, as well as the figures to support findings. Though I would recommend adding to the description of, Figures 5, 6, & 7, to state where the reported performance comes from (e.g., averaged across, etc.). Additionally, the axis labels in Figure 9 are nearly impossible to read. Though not extremely important, it would be helpful to provide a figure with higher resolution so that they can be read.

Conclusions are well stated and impact of the novel model is described.

1) When testing with the other models, did you use the same 5-set cross-validation approaches across all models? Please provide more details to confirm the similarity in the approaches so that it is easier to compare the various models and to be confident that differences in consistency of the predicted scores are due to the models themselves, not just the other processes/steps.

2) In line 111, is this the dataset that you refer to later as the Isola dataset? If so, please label it here on the first time you introduce it. If not, what makes this different? In line 328, the same comment -- is this the Isola dataset?

3) In the supplementary material you provide the link to the LaMem dataset, however, can you also add how to access the Isola dataset?

Additional comments

Minor writing suggestions
1) A few instances of unnecessary capitalization throughout, for example:
Line 84 “Image Memorability”
Line 440 “Computer Vision tasks”
Line 491 “Image Aesthetics”
2) Line 202, the use of ResNet-50 is inconsistent, please use the same phrasing throughout so it is easier for the reader to follow
3) Line 276 “it’s” should be “its”
4) Line 418, the word “close” appears twice
5) Line 161 “ResMet-Net” should be “ResMem-Net”
6) 508 “per-trained” model should be “pre-trained model”

---

## Round 0.2 · Minor Revisions

The manuscript has improved from the previous version. The authors should incorporate the reviewers' comments to improve the quality of the manuscript.

Reviewer 1 ·

Basic reporting

.

Experimental design

.

Validity of the findings

.

Additional comments

.

Reviewer 2 ·

Basic reporting

no comment

Experimental design

no comment

Validity of the findings

no comment

Additional comments

This research work "ResMem-Net: Memory based deep CNN for image memorability estimation" is proposes. The research domain addressed in this research paper is novel, and the results are satisfactory. But it needs the below improvements:

1. Reduce unnecessary text from the paper.
2. The authors should discuss more on deep CNN.
3. The Experiment discussion should be written more in-depth, precise, and concrete, such as what questions were resolved? How can the proposed method solve these problems? The most recent works should be discussed in the related work section.
5. The objectives of this paper need to be polished. The contribution list should be polished at the end of the introduction section, and the last paragraph of the introduction should be the organization of the paper.
6. Reduce the size of the introduction.
7. Contributions at the end of the introduction section should be polished.
8. Relevant literature reviews of the latest similar research studies on the topic at hand must be discussed.
9. The quality of the figures is not good.
10 Don’t use “Its,” ‘I, your’ and any informal text in the paper.
11. There are some grammar and typo errors.

Reviewer 3 ·

Basic reporting

NA

Experimental design

NA

Validity of the findings

NA

Additional comments

NA

·

Basic reporting

Good

Experimental design

Good

Validity of the findings

Good

Additional comments

All corrections are addressed by authors.

Reviewer 5 ·

Basic reporting

I appreciate the authors' diligence and am satisfied with how they have addressed a majority of my concerns and incorporated my feedback and suggestions to improve the manuscript. I do have two minor concerns (described below) that I think could be addressed to further improve the manuscript. First, while I see how the authors addressed my concerns and provided additional details to help clarify their intention with the use of the term "image memorability", I still have a remaining question as to whether they are trying to portray that image memorability is a property of the image or more a reflection of the individual who is viewing the image? I feel like the authors could expand this around lines 112 and 113, perhaps by adding a bit more to the sentence in 114 and 115 to help make this clearer.

Experimental design

I would ask the authors to provide a citation/support for the claims made in line 854. Additionally, it would be helpful if they could add some statistics pertaining to the findings of violence and image aesthetics relating to memory, lines 854 - 856. It would be helpful if they followed the same format that they had used previously when reporting about the impact of the presence of humans on memorability.

Validity of the findings

No further recommendations or suggestions. Nicely done!

Additional comments

Minor writing suggestions
1) Line 220, “Suport” should be “Support”
2) Line 492-493, missing closing parenthesis “)”

---

## Round 0.3 · accepted · Accept

The manuscript has improved significantly in two rounds of review. The authors should proofread the manuscript before submitting the final files.

Reviewer 2 ·

Basic reporting

'no comment

Experimental design

'no comment

Validity of the findings

'no comment

Additional comments

'no comment

Reviewer 5 ·

Basic reporting

No comments

Experimental design

No comments

Validity of the findings

No comments

Additional comments

No further recommendations or suggestions. Nicely done!